# Variation in Shoot, Peduncle and Fruit Growth of *Lagenaria siceraria* Landraces

**DOI:** 10.3390/plants12030532

**Published:** 2023-01-24

**Authors:** Lungelo Given Buthelezi, Sydney Mavengahama, Julia Sibiya, Nontuthuko Rosemary Ntuli

**Affiliations:** 1Department of Botany, Faculty of Science, Agriculture and Engineering, University of Zululand, KwaDlangezwa 3886, South Africa; 2Food Security and Safety Area, Faculty of Natural and Agricultural Science, North-West University, Mmabatho 2745, South Africa; 3School of Agriculture, Earth and Environmental Sciences, University of KwaZulu-Natal, Pietermaritzburg 3209, South Africa

**Keywords:** correlation, variability, harvested shoots, peduncle, fruit

## Abstract

*Lagenaria siceraria* (Molina) Standley is a prominent food source as almost all its plant parts are edible. However, no studies have recorded the changes in shoots, peduncles and fruits during its growth. Hence, this study aimed to record changes in shoot traits and relate the peduncle to the fruit traits of *L. siceraria* landraces across different growth stages. Changes in shoots, peduncles and fruits during growth were compared within and among landraces using analysis of variance, correlation, principal component analysis, cluster analysis and heritability estimates. Almost all landraces had harvestable shoots at 42 days after sowing. Peduncles became shorter and wider as the fruits elongated. Shoots, peduncles and fruits correlated positively with each other. The informative principal components had a total variability of 84.488%, with a major contribution from shoot traits. The biplot and dendrogram clustered landraces with similar growth habits and the harvestable shoot and fruit attributes into three clusters, but KRI and NSRC formed singlets. Shoot width (60.2%) and peduncle length (55.2%) had high heritability estimates. The general low heritability estimates and genetic advances indicated the presence of non-additive gene action. This study is the first report on changes in harvested shoots and the relationship between peduncles and fruits during growth.

## 1. Introduction

*Lagenaria siceraria* (Molina) Standley, commonly known as the bottle gourd, is a member of the Cucurbitaceae family [1]. It is a herbaceous vining crop identified by its andromonoecious cream-to-white-colored flowers different from other cucurbits [2]. The plant is a cost-effective food source that is often labeled as a poor man’s crop, but it requires low cultivation inputs and thrives under extreme growing conditions [3,4]. *Lagenaria siceraria* is widely used for food, medicinal, utensils and decorative purposes, which influences its preservation by subsistence farmers and rural communities [5,6]. 

The growth of shoots [7] and fruits [8] is facilitated by an increased number and/or volume of cells. The number of cells is correlated to the rate and duration of cell division, and a larger cell volume results from accelerated cell expansion [8]. The primary transportation of photo-assimilates from vegetative modules through growing shoots to flowers and fruits is facilitated by the peduncle, a flower stalk that bears pistillate flowers in cucurbits [9]. At anthesis, the peduncle diameter increases significantly from 0.9 mm at 0 days after anthesis (0 DAA) to 3.5 mm at 45 DAA along with the phloem and xylem area that is responsible for the translocation of assimilates [8]. An increased translocation area leads to a high net photosynthetic rate resulting in the formation of vigorously growing plants [7] and significantly larger cucurbit fruits [8]. 

Correlation represents the relationship between variables or responses that may differ or be similar to each other, where the extent of the relationship is expressed in values that range between “−1 and 1” [10,11]. Large sample sizes are important to use in constructing a correlation matrix to achieve reliable statistical significance among the studied traits [12]. Furthermore, principal component analysis is a simplified representation of a dataset that accounts for variability among the investigated variables [13]. Principal components with eigenvalues of ≥1 are selected for data interpretation because such values signal the presence of variation among traits and are traditionally considered worth analyzing [14]. In variation studies, cluster analysis is conducted in the form of a biplot and dendrogram. A biplot facilitates a visual representation of correlations between different accessions based on their morphological attributes while acknowledging the environmental and genetic impact on their display [15]. Moreover, a dendrogram is a tree-like structure that visualizes the sequence of hierarchical clustering based on similarities or dissimilarities among a set of varying individuals [16]. 

Genetic parameters such as phenotypic and genotypic coefficients are used to measure the variation among different genotypes [17]. The phenotypic display of a plant is bound to the genetic value, environment, and interaction of both factors [18]. Heritability measures the impact of the genetic makeup on the expression of a desired attribute [17]. A heritability estimate also determines the selection, such that it displays the quantifiable representation of a trait due to genetics or the environment that can be further utilized in crop advancements such as hybridization [18]. Crop advancement greatly depends on the amount of genetic variability that is present and how much of it is heritable [19]. A heritability estimate close to 100% suggests a stronger influence that genetics has on expressed attributes among different germplasms, whereas an estimate closer to 0% suggests that most of the variation is not driven by genetics [18,19]. Furthermore, broad-sense heritability measures the phenotypic diversity due to genotypic values that may result from dominance and epistatic gene action [19]. On the other hand, narrow-sense heritability represents the genetic variation that results from additive genetic values [19]. Growth traits of *Lagenaria siceraria* fruits such as the size, mass, length, width and yield of fruits are the most heritable traits [20]. These traits have high genotypic coefficient values, which advocate the validity of selecting favorable genotypes based on fruit growth traits [2,21].

To date, no study has investigated the changes in the size of sequentially harvested shoots, and the correlation between the peduncle (length and width) and fruit (length, width and mass) traits during growth. Therefore, the aim of this study was to record changes in the traits of harvested shoots and relate the peduncle to the fruit traits of *L. siceraria* landraces across different growth stages.

## 2. Results

### 2.1. Shoot Traits of Lagenaria siceraria Landraces

Almost all *L. siceraria* landraces from the KwaZulu-Natal and Limpopo provinces (Table 1) had harvestable shoots at 42 days after sowing (DAS) except for landrace BG-19, whose shoots were ready for harvest at 49 DAS (Table 2). The majority (88%) of landraces had their shoots ready for harvest every seven days from harvest commencement. However, the shoots of landraces BG-70 and NqSC were harvestable at 14-day intervals, particularly in the early stages of growth. The traits of harvested shoots varied significantly (*p* < 0.05) among landraces within the same harvest day and within each landrace at different stages of growth.


Among those landraces that had harvestable shoots at 42 DAS, landraces DSI, KRI, KSC, KSP and NqSC had longer shoots than landrace ESC. Again, DSI, KRI, NRC and NSRP had longer shoots than KSC at 49 DAS. Furthermore, at 56 and 63 DAS, landrace KRI continued to produce longer shoots than most of the other landraces, with few exceptions. A comparison of shoot lengths in each landrace recorded that longer vine were harvested at 56 than at 42 DAS in landrace BG-27, as well as at 63 than at 42 DAS in BG-31, BG-81, BG-100/GC, ESC and NRC. Again, shorter vines were harvested at 49 than at 42 DAS in KSC, as well as at 49 and 56 than at 42 DAS in KSP. However, in other landraces, harvested shoots from each landrace had similar lengths at all stages of growth. 

Shoots of all landraces harvested at 42 and 63 DAS had similar widths (Table 2). Landraces KRI and KSP at 49 DAS, and BG-31 and KRI at 56 DAS, had thicker shoots than other landraces. The shoots of BG-31 and KSP harvested at 56 and 49 DAS, respectively, were thicker than at other stages of growth within each landrace. Again, KRI produced wider shoots at 49 than 42 DAS. Fresh shoots harvested from landraces DSI, KRI, NqSC and NRC were heavier than those of other landraces at 42 DAS (Table 2). However, the shoots of landraces DSI and KRI were only heavier than the shoots of landraces BG-26, BG-27, BG-81, KSC and KSP at 49 DAS. Landrace KRI documented the heaviest shoots at 56 DAS, but they were comparable to those of BG-27. Again, at 63 DAS, the higher shoot mass of KRI was similar to that of landraces BG-100/GC, NqSC, NRC and NSRP. Landraces BG-27 and NSRP recorded heavier shoots at 63 than at 42 DAS. However, the shoots of landraces DSI and NqSC were heavier at 42 DAS than at 56 DAS for both, as well as at 63 DAS for DSI. Alternatively, other landraces maintained a similar shoot fresh mass across all growth stages.

The dried shoots of landraces DSI, KSC, KSP and NqSC were heavier than those of BG-26 and NSRP at 42 DAS (Table 2). Both landraces DSI and NRC had heavier dried shoots than BG-27 at 49 DAS. Moreover, landrace KRI produced the heaviest dry shoots at 56 DAS, but this was similar to BG-100/GC. However, the mass of landrace BG-100/GC dried shoots was only higher than that of BG-27, BG-70, BG-80 and ESC. At 63 DAS, landraces BG-100/GC, KRI and NRC had heavier dry shoots than landraces BG-27 and NSRP. The mass of landrace DSI dried shoots was higher at 42 DAS than at 56 and 63 DAS (Table 2). Landrace KRI recorded heavier dried shoots at 56 DAS than at other stages of growth. Again, the dried shoots of landrace KSC were heavier at 42 DAS than at 49 and 63 DAS. Furthermore, landrace NqSC recorded a higher mass of dried shoots at 42 DAS than at other stages of growth, including the non-harvestable shoots at 49 DAS. In landrace NSRP, a higher shoot dry mass was recorded at 49 than at 63 DAS. 

Landrace NRC had the highest shoot moisture content, which was similar to that of DSI, KRI and NqSC at 42 DAS (Table 2). However, the moisture contents of these landraces (DSI, KRI and NqSC) were higher than those of landraces BG-27, BG-31, BG-70, BG-81, BG-100/GC, ESC and KSC. Moreover, landraces BG-26 and KSP had the lowest shoot moisture contents among landraces with harvestable shoots at 42 DAS. Landraces DSI, KRI and NRC produced shoots with higher moisture contents than all landraces at 49 DAS. Landraces KRI and NRC continued to produce shoots with high moisture contents, including landraces BG-27 and BG-80, at 56 DAS. These landraces had significantly higher shoot moisture contents than BG-100/GC and KSP. Furthermore, landrace BG-80 continued to produce a significantly higher shoot moisture content than KSP, though both landraces did not vary when compared with all the other landraces at 63 DAS. The shoots of landraces BG-19, BG-26, BG-70, BG-81, BG-100/GC, KSC, KSP and NSRP were sappier at 63 DAS than at other growth stages (Table 2). In shoots of landraces BG-24 and BG-31, higher moisture content was recorded at 63 DAS than at 42 and 56 DAS. Moreover, landraces BG-27, BG-80 and ESC had higher shoot moisture contents at 63 DAS than at 42 and 49 DAS. Furthermore, the shoots of landraces DSI and NqSC were sappier at 63 than at 56 DAS.

### 2.2. Dynamics in the Development of Peduncles, Intact and Harvested Fruits of Lagenaria siceraria Landraces

Petals of *L. siceraria* landraces wilted from the periphery towards the core within approximately six hours after pollination. The peduncles curved to orientate the developing fruits towards the ground. Landraces with curvilinear or cavate (KSC, NSRC, NqSC, ESC, BG-24, BG-26, BG-27, BG-31, BG-80 and BG-81), isodiametric (KRI, DSI and BG-19), and pyriform or pear-shaped (KSP, NSRP and BG-70) fruits sit on the ground with their apex (fruit scar). However, the cylindrical (oblong) fruits of NRC and BG-100/GC landraces lie on their side.

The majority (88%) of landraces produced measurable intact fruits from 0 to 5 days after anthesis (DAA) in vining shoots (Table 3). Landraces NqSC, NRC and NSRC had high fruit abortion rates, and, thus, fruit traits were not documented during all stages of growth. Furthermore, landrace NSRC was not included in the shoot traits because of its retarded growth.

The peduncle lengths at 0 DAA varied significantly, wherein the longest peduncles in landrace BG-100/GC were similar to those of BG-27 (Table 3). Landrace BG-24 also had longer peduncles than landraces BG-19, BG-26, BG-80 and KSP. Again, landraces BG-27 and BG-100/GC maintained longer peduncles at 1 DAA compared with landraces BG-19, BG-26, BG-31, BG-70, BG-80, BG-81, DSI and KSP. Additionally, more variations were observed at 2 DAA, such that landraces BG-24, BG-27 and BG-100/GC recorded longer peduncles than BG-19, BG-26, BG-70, DSI and KSP. Furthermore, landrace BG-24 had longer peduncles at 3 DAA than landraces BG-19, BG-70, DSI, KRI and KSP. At 4 DAS, landrace BG-24 maintained its longer peduncles over landraces BG-31, BG-70, DSI, KRI and KSP. At 5 DAA, landrace BG-24 still produced longer peduncles compared with BG-19, BG-26, BG-31, BG-70, BG-80, DSI, KRI and KSP.

A total of 15 landraces had measurable peduncle and fruit traits soon after anthesis, and 87% of them showed a reduction in peduncle length during growth (Table 3). Peduncles of landraces BG-24 and BG-26 were shorter at 5 DAA than at 0 DAA. Landraces BG-27 and DSI recorded shorter peduncles from 2 to 5 DAA than at 0 DAA. Furthermore, landraces BG-31, BG-81, ESC, KSC and KSP produced shorter peduncles at 4 and 5 DAA than at 0 DAA. In landrace BG-100/GC, the shortest peduncles were recorded at 4 and 5 DAA, and peduncles measured at 1DAA were also shorter than those at 0 DAA. Landrace DSI had shorter peduncles at 2, 3, 4 and 5 DAA than at 0 DAA. Again, landrace KRI produced longer peduncles at 3, 4 and 5 DAA than at 0 DAA. 

Peduncle widths recorded at 0 and 1 DAA were similar among all landraces that produced fruits (Table 3). However, at 2 DAA, landrace KSP had wider peduncles than those of BG-27, BG-80 and BG-81. Landrace KRI recorded wider peduncles than BG-27, BG-81 and ESC at 3 DAA. Furthermore, landrace KSP had wider peduncles than landraces BG-24, BG-80 and BG-81 at 4 DAA. Moreover, at 5 DAA, landraces BG-19, BG-26 and KSP had wider peduncles than BG-24, BG-27, BG-31, BG-80, BG-81 and ESC.

Peduncle widths increased as they grew from 0 to 5 DAA, such that thicker peduncles were recorded at 5 DAA when compared with other stages of growth (Table 3). Landraces BG-19 and BG-26 documented wider peduncles at 5 DAA than at other stages of growth. Thicker peduncles were recorded in landraces BG-24, BG-100/GC, DSI, KRI, KSC and KSP at 5 DAA than at 0 and 1 DAA. Landraces BG-70 and NSRP had wider peduncles at 5 DAA than at 0–2 DAA. Peduncles recorded at 5 DAA for landraces BG-80 and BG-81 were wider than those at 0 DAA. Again, peduncles recorded at 4 DAA in landrace ESC were wider than those at 0–3 DAA. 

Landraces did not vary in their fruit length at 0 DAA (Table 3). At 1 DAA, landrace KSP recorded longer fruits than BG-19. Landrace ESC produced longer fruits than BG-19, BG-26, BG-100/GC and KRI at 2 DAA. Landraces ESC, KSC and KSP had longer fruits than BG-19 at 3 DAA. Longer fruits were recorded in landraces BG-27, KSP and NSRP than in BG-19 and BG-100/GC at 4 DAA. The longest fruits recorded at 5 DAA in landrace BG-27 were similar to those of NSRP. Again, at 5 DAA, landrace DSI produced longer fruits than BG-19, BG-26, BG-70, BG-80, BG-81, BG-100/GC and KRI. Most landraces (69.23%) with measurable fruits had an increase in fruit length from 0 to 5 DAA (Table 3a). Landraces BG-24, ESC and KSC recorded longer fruits at 5 DAA than at 0 DAA. Furthermore, landrace BG-27 had the longest fruits at 5 DAA, and fruits measured at 4 DAA were also longer than the ones at 0 DAA. Fruits of landrace NSRP were longer at 5 DAA than at other growth stages. Fruits of landrace DSI at 5 DAA and of KSP at 4 DAA were longer than those recorded from 0 to 3 DAA and 0 DAA, respectively.

Landraces with fruits at 0, 1 and 2 DAA did not vary in their fruit widths (Table 3). Broader fruits were recorded in landrace DSI than in BG-27, BG-70 and NSRP. At 4 DAA, landrace DSI had wider fruits than almost all landraces except for the fruits of landraces BG-26 and BG-31. Furthermore, landraces BG-31, DSI and NSRP produced wider fruits than BG-70, BG-80, BG-100/GC, ESC, KRI and KSP at 5 DAA. Fruit expansion within 5 days after anthesis was evident in the majority of landraces (76.92%) with measurable fruits (Table 3). Fruits recorded at 5 DAA were wider in landraces BG-19, BG-24 and BG-81 than at 0 DAA, and in BG-19 and BG-81 than at 1 DAA. Furthermore, landraces BG-26, BG-31 and KSC produced wider fruits at 5 DAA than from 0 to 3 DAA.

Landraces BG-24, DSI, NqSC and NRC had longer peduncles than BG-27, BG-31, KSP, NSRC and NSRP (Table 4). Peduncles produced by landrace KSP were longer than those of the other landraces except BG-24, ESC, NRC and NSRC at 14 DAA. Landrace KSP continued to produce longer peduncles than BG-31, ESC, KRI and NSRP at 21 and 28 DAA, as well as NSRC only at 28 DAA. Only landraces KRI, KSP and NSRC had variations in peduncle length when they grew from 7 to 28 DAA. Shorter peduncles were recorded at 28, 7 and 28 DAA than at 7, 14 and 14 DAA for landraces KRI, KSP and NSRC, respectively. 

### 2.3. Correlations between L. siceraria Shoot, Fruit and Peduncle Traits at Different Growth Stages

Almost all shoot traits had a positive correlation with each other at different growth stages except for a negative correlation between shoot dry mass and shoot moisture content at 56 and 63 days after sowing (DAS) (Table 5a). Peduncle length correlated negatively with peduncle width and fruit width at 2–5 and 4–5 DAA, respectively (Table 5b). Peduncle width correlated positively with fruit length at 4 DAA and also with fruit width at 0 and 3 DAA. A positive association was recorded between the length and width of fruits at different stages of growth, except at 0 and 4 DAA. All vegetative traits had a positive correlation with each other (Table 5c). Again, a positive correlation was evident between the length, width and mass of fruits. 

### 2.4. Principal Component and Cluster Analyses

The first three informative principal components (PC1, 2 and 3) were responsible for 86.488% of the cumulative variability, with each principal component having an eigenvalue greater than 1.0 (Table 6). The first principal component (PC1), responsible for 47.233% of the total variation was strongly positively correlated with the shoot length, shoot width, shoot fresh mass, shoot dry mass and shoot moisture content. The second principal component (PC2), with 24.681% of the total variability, was positively associated with the peduncle width, fruit mass, fruit length and fruit width. The third principal component (PC3) responsible for 14.574% of the total variation was positively correlated with the peduncle length but negatively correlated with the peduncle width.

In a biplot, all shoot traits correlated positively with the first principal component but negatively with the second principal component (Figure 1). Furthermore, the length, width and mass of fruits had a positive association with both components. However, peduncle width and length had a positive correlation with only PC2 and PC1, respectively. 

Landraces were grouped into three clusters, whereby landraces KRI and NSRC each formed a singlet, in a biplot. The first cluster consisted of landraces DSI, KSC, KSP and NSRP. The second cluster included landraces ESC, BG-24, BG-27, BG-31, BG-100/GC and NRC. The third cluster consisted of landraces BG-19, BG-26, BG-70, BG-80, BG-81 and NqSC.

A hierarchical cluster analysis grouped landraces into five major clusters (clusters I–V) (Figure 2). Cluster I consisted of landrace NSRC as a singlet, and cluster II was composed of BG-19, BG-70 and NqSC. Landraces BG-27, BG-100/GC, ESC and KSC were in cluster III, whereas KRI and NRC were in cluster V. All other landraces were grouped in cluster IV.

#### Genetic Parameters

The genetic parameters varied among the shoot, peduncle and fruit traits (Table 6). The highest genetic variance was documented for the length of harvested fruits (7771.7) while the lowest was recorded for shoot width (0.1). Environmental variance ranged from 5.6 × 10^−3^ (peduncle width of intact fruits) to 4156.0 (shoot length). Shoot length recorded the highest phenotypic variance (11,452.5) while shoot width had the lowest (0.1). The grand mean ranged from 312.1 (shoot length) to 0.4 (peduncle width of intact fruits).

Phenotypic and genotypic coefficients of variation were the highest (˃50%) for the length of harvested fruits (65.9% and 63.3%), while the peduncle width of harvested fruits had the lowest (1.7% and 1.5%). The shoot width reported the highest heritability estimate (60.5%), while shoot moisture content had the lowest (2.0%). The highest genetic advancement of 0.5% was recorded for shoot moisture content and fruit length, width and mass of harvested fruits while the lowest (0.3%) was recorded for shoot width and peduncle length of harvested fruits.

## 3. Discussion

### 3.1. Shoot Growth of Lagenaria siceraria Landraces

The common initiation of *Lagenaria siceraria* shoot harvest at 42 days after sowing (DAS), except for BG-19 harvested at 49 DAS (Table 2), can be related to the variation in the shoot growth rate among landraces. The delayed formation of harvestable shoots in landrace BG-19 was probably caused by the production of a high auxin concentration, which inhibits the formation of lateral branches by acting in the biosynthesis pathway of strigolactones and cytokinins phytohormones, which have an antagonistic effect on shoots and tertiary growth in vegetative traits [25,26]. 

Differences of seven and fourteen days in shoot harvest frequency among *L. siceraria* landraces (Table 2) can be attributed to the differences in auxin activity, influx of photo-assimilates that are responsible for branching (regenerative ability), and rate of shoot growth [27,28]. The frequency of shoot tip harvest redirects high auxin concentrations to the lower parts of a plant and causes an increase in the plant’s physiological activities and the influx of carbohydrates such as sucrose [28]. Sucrose is a carbon source that is utilized at sink sites (vegetative modules, axillary buds and growing fruits) to supply the energy required for plant biomass production [27] and, also, to regenerate new shoots [29] for the next harvest, particularly in the current study. Although shoot harvesting is beneficial, it can injure plants and make them prone to environmental stress, pests and fungal attacks, which lowers plant hardiness and productivity [30]. The seven- and sometimes fourteen-day shoot harvest intervals in the current study (Table 2) were different from the ten-day interval harvest period among *L. siceraria* landraces in Zimbabwe [31]. The extended time between harvest intervals allowed landraces in the comparative study more time to accumulate more photosynthetic assimilates, and subsequently promoted the development of longer shoots [13].

The seven-day shoot harvest frequency in most of the studied landraces (Table 2) is evidence of the high growth rate in the *L. siceraria* species in general but also a delayed growth rate in landraces BG-70 and NqSC (with 14-day intervals). These landraces with delayed growth rates also produced some of the shortest and narrowest shoots across different growth stages (Table 2). Shoot growth over time is primarily driven by gibberellins, which cause cell proliferation and elongation in shoots by stimulating ribonucleic acid and protein synthesis for the growing shoots and the entire plant [32]. As a result, an increase in photosynthetic activity enhances shoot length and biomass [32]. *Lagenaria siceraria* landraces with early and frequent shoot harvests have productivity attributes that are ideal in agriculture because they enable researchers and farmers to select high-yielding genotypes and ensure market availability and sustainability [26,27]. 

### 3.2. Peduncle and Fruit Sizes of Lagenaria siceraria Landraces at Different Growth Stages

The wilting of petals soon after pollination in the studied *L. siceraria* landraces was probably facilitated by ethylene. Pollinated stigmas produce ethylene that moves to the petals [33] and induces the production of hydrolytic enzymes, which degrade the carbohydrates, proteins, lipids and nucleic acids in the petals and channel them to the developing ovary [34]. These enzymes also digest the photosynthesized sucrose into readily available replenishment molecules, such as hexose phosphate, bolstering the mass flow of assimilates to the sink sites [35]. 

The bending of peduncles can be associated with the activity of the phytohormones auxin and ethylene. During peduncle bending, auxin is concentrated on the concave (elongating) side [36] and a random orientation of cellulose microfibrils is induced by auxin and ethylene [37]. The production of G-fibers (gelatinous or tensile wood fibers) can also cause the curving of *L. siceraria* peduncles with developing fruits in a similar way, by increasing the tensile force as in woody plants [38,39,40]. The different orientation in fruits towards the ground can be attributed to the sedimentation of gravity-sensing starch granules, statoliths [35,41], which could be the by-product of hydrolyzed carbohydrates from senesced petals [29,41]. Statoliths are membrane-bound starch molecules (amyloplasts) in gravity-sensing cells that can be found in shoots and roots [35]. The basis of the theory is that statoliths are redirected to the ground due to the association with the gravity-sensing cells, which direct the growth direction [35]. According to general observations, the orientation of amyloplasts towards the terminal end of a sink organ is the primary component of gravity-sensing that triggers downstream signaling and generates biochemical and physiological responses in the responsive plant tissues [35]. Therefore, during peduncle curving, statoliths in curvilinear, isodiametric and pear-shaped fruits will sediment at the tip of the fruit, whereas in oblong fruits, statoliths will sediment randomly, as assisted by auxin [36]. In *Cucumis sativus*, an asymmetric distribution of auxin regulates the development of fruit shape, whereby in curved fruits, high auxin concentration is recorded on the convex rather than concave side, but similar auxin distribution is recorded on both sides of straight fruits [36]. 

In the present study, peduncle length decreased from 111.5 to 17.1 mm at 0 to 5 DAA, and from 104.0 to 27.27 mm at 7 to 28 DAA, whereas peduncle width increased from 3.1 to 7.3 mm at 0 to5 DAA and 0.36 to 0.95 mm at 7 to 28 DAA. The decrease in the peduncle length can be associated either with changes in the orientation of cellulose microfibrils and a decrease in turgor pressure within cells [42], or with the collapse of parenchyma cells and the production of G-fibers [38]. Peduncle elongation, which is necessary for flower exposure to pollinators [13] prior to anthesis, is facilitated by gibberellic acid hormones through the horizontal orientation of cellulose microfibrils and an increase in cell turgor pressure that enables cell expansion [37]. However, hormonal changes soon after anthesis probably trigger the actions of auxin and ethylene, which promote radial micellation (the vertical orientation of cellulose microfibrils) in the cell walls [37], whereby the cell increase (expansion) sideways will result in peduncle thickness [13]. The increase in abscisic acid and ethylene can cause a decrease in the turgor pressure of the same cells [43] and result in shorter peduncles. Peduncle shortening can also result from the collapse of vascular and cortical parenchyma cells, which happens during the contraction of parts such as stems and roots in selected plants [38]. However, the wrinkled surface that is morphologically obvious in contractile plant parts [38] was not evident in *L. siceraria* peduncles except in twisting and curving. The production of G-fibers can enhance the tensile strength and facilitate the bending and contraction of *L. siceraria* peduncles during growth, as evidenced in the anatomy of woody plants and cycads [38,39]. Studies on anatomical differences and hormonal changes within developing peduncles as well as the sedimentation of statoliths in developing *L. siceraria* fruits are required. 

In pedunculated cucurbits such as *Cucumis sativus*, fruit growth is attained through cell expansion and progressive maturation until ripening, which resembles a sigmoid growth habit [44,45,46]. Fruit cell number and size are key limiting factors in determining the fruit size and shape, whereby fruit growth is maintained by an accelerated cell division from 0 to 6 DAA, and, thereafter, an exponential enlargement stage from 6 to 15 DAA [45,47]. Such growth and enlargement are largely controlled by phytohormones such as auxin, gibberellins, cytokinins, abscisic acid and ethylene [46]. Before anthesis, fruit shape and size are correlated with the shape and size of developing inferior ovaries [48,49]. A related study on *C. sativus* recorded an increase in the ovary size pre-anthesis, arguably due to the presence of endogenous phytohormones in the ovules as early as six days before anthesis (DBA) [46,48]. This phenomenon drives the orientation and magnitude of cellular division, subsequently impacting the fruit shape and size of cucurbits [47].

### 3.3. Correlation

A positive correlation among all shoot traits across all growth stages (from 42, 49, 56 to 63 DAS) (Table 5a) suggests that changes in shoot traits are dependent on each other irrespective of differences in growth stages (harvest periods) [11]. Similarly, in *C. sativus*, shoot mass correlated positively with shoot length as the plants aged from 14, 21 to 28 days after planting (DAP) [50]. 

The negative correlations between dry mass and moisture content of shoots at 56 and 63 DAS (Table 5a) are indicative of the reduction of sappiness in *L. siceraria* shoots as the plants grow. Before anthesis, all the photosynthetic by-products are concentrated towards vegetative growth, and soon after anthesis begins, vegetative growth is temporarily suspended to accommodate the developing fruits from 52 DAS until plant senescence [27,51]. As a result, the sappiness of shoots is reduced due to the accumulation of secondary shoot tissues (cellulose, hemi-cellulose and lignin), which promote shoot dry mass (shoot biomass) as the plant is growing [27,30].

The negative correlation of peduncle length with peduncle width from 2 to 5 DAA and with fruit width from 4 to 5 DAA (Table 5b) can be attributed to the shortening of peduncles as they become thicker and as fruits widen during growth (Table 3). Peduncles, as source organs that are photosynthetically active, redirect their metabolized by-products to the growing fruits, and this process results in a decrease in their length [13,52]. During the growth of sink organs (developing fruits) the mass flow of assimilates from source modulates to sink organs also increases [52]. However, the widening of peduncles during the growth of the studied landraces can be related to the thickening of parenchyma cells, fibers, vascular bundles and pith parenchyma [13].

### 3.4. Principal Component Analysis

The presentation of variability among the studied traits in the current study was simplified with a principal component analysis (Table 6) [13], wherein principal components with eigenvalues of ≥1 were considered for analysis [14]. The eigenvalues of 4.723, 2.468 and 1.475 for the first, second and third principal components of this study (Table 6) indicated high variability among landraces based on the studied traits and were thus considered worth analyzing [14]. In the current study, the first principal component that accounted for 43.233% of the total variability was positively defined by the length, width, fresh mass, dry mass and moisture content of shoots. Similarly, PC1 of *L. siceraria* genotypes from India, with an eigenvalue of 3.13, accounted for 24.11% of the variability, which was also contributed to by shoot internodal length and total vine length [13].

Furthermore, PC2 was responsible for 24.681% of the variability and was associated with peduncle width, as well as the length, width and mass of fruits in the current study. In corresponding results for germplasm from India, the second principal component had an eigenvalue of 1.64 responsible for 12.64% of the variation in the featured fruit length and fruit width [13]. The difference in variability among the studied traits can be attributed to the sequential (growth) measurement of traits in the current study versus the once-off measurements in *L. siceraria* accessions in India [53]. Furthermore, the differences in the measured traits between these studies may contribute to the variation in the results, such that the only common traits were peduncle length as well as the length, width and mass of fruits.

The current study indicates that shoot traits are principal determinants of variation among *L. siceraria* landraces (significantly correlated with PC1), followed by fruit traits (associated with PC2). However, previous studies indicate fruit traits as the most limiting characteristics that are responsible for variability among *L. siceraria* landraces [13,53]. The possibility for shoot traits to give high variability may be associated with the measurement of only the harvested three-leaved shoots (tips) rather than the main vine as in other studies, as well as the data on developing intact and harvested fruits in this study versus those of mature fruits in other studies. 

Although PC3 had the least variability percentage (1.457%), it clarified a contrast between the length (r = 0.850) and width (r = −0.672) of peduncles (Table 6). This reflects the reduction in length but widening of the peduncles as they grow, particularly from 0 to 5 DAA (Table 3), and the negative correlation among these traits (Table 5b). 

### 3.5. Cluster Analysis

The grouping of landraces with similar growth trends in shoots and fruits in a biplot (Figure 1) concurred with the clustering of *L. siceraria* landraces with similar root characteristics in Chile and Mozambique [54]. This was evidence of correlation among different *L. siceraria* landraces based on their morphological attributes, as facilitated by the biplot [15]. 

In the biplot (Figure 1), the grouping in the first cluster was primarily based on smooth-textured landraces, which also produced the longest, widest and heaviest fruits (Table 4). These landraces are ideal for the substance of human dietary requirements and consumption due to their high-yielding characteristics. Landraces in the second cluster had harvestable shoots in all growth stages that were longer, wider and heavier as well as the longest peduncles. These landraces have, therefore, a potential for sustainable shoot harvesting, which will contribute towards food security [55]. Landraces in the third cluster were associated based on lower shoot regeneration rates, high fruit abortion rates and small-sized fruits. Such landraces require studies on improving their growth and yield capacity. Landrace KRI, which was a singlet, had harvestable shoots in all growth stages which were the longest, widest and heaviest. The other singlet, landrace NSRC, was characterized by the most retarded shoot growth (as it was excluded in the shoot data) and a high fruit abortion rate. Thus, this landrace (NSRC) is identified as the chief germplasm target for future research on growth and yield.

Landrace NSRC was also a singlet in the dendrogram, whereas KRI was grouped with NRC (cluster V). The association of KRI and NRC can be based on their rough fruit texture and small-sized fruits. Again, the grouping of landraces BG-19, BG-70 and NqSC in cluster II of the dendrogram relates to their association in the biplot (cluster III). Landraces in the third cluster of the dendrogram had longer peduncles at anthesis that had a quicker reduction rate within five days after anthesis. Cluster IV had a mixture of landraces with vigorous shoot growth (BG-26 and BG-31), high fruit abortion rates (BG-24, BG-80 and BG-81), and with big and heavier fruits (DSI, KSP and NSRP).

### 3.6. Genetics Parameters

The higher genotypic variance and genotypic coefficient of variation than the environmental variance and environmental coefficient of variation in almost all traits, except for shoot width and peduncle length of harvested fruits (Table 7), indicate that the phenotypic variation recorded in the studied landraces is primarily caused by genetic differences. However, the higher values of phenotypic variance and phenotypic coefficient of variation than the genotypic variance and genotypic coefficient of variation explain some minor environmental impact on the variation in the studied traits [20].

The high heritability values recorded in shoot width (60.5) and peduncle length of harvested fruits (55.2) suggest that the environment had a low impact on their expressions [19]. However, their expressions were not strongly influenced by genetics because their estimates were far lower than 100% [19]. Low–moderate heritability estimates coupled with a low genetic advancement in the current study indicate that the majority of investigated traits were governed by non-additive gene action, which imposes difficulties in their selection for breeding purposes [17]. 

## 4. Materials and Methods

### 4.1. Germplasm Sourcing

Eighteen landraces of *Lagenaria siceraria* from different agro-climatic regions in Northern KwaZulu-Natal and Limpopo, South Africa were investigated (Table 1). Landraces from KwaZulu-Natal were named according to their area of origin represented by the first letter, fruit texture represented by the second letter, and fruit shape represented by the third letter (Table 1). Landraces from Limpopo were named by previous investigators based on their entry number and distinguished by their fruit and seed traits [3,22,23,24]. Seeds of the landraces were collected from Ga-Phasa (23.4057° S, 29.1557° E), Kgohloane (23.4739° S, 29.2213° E), Khangelani (29.0106° S, 31.2211° E), Moletjie-Mabokelele (23.4514° S, 29.1713° E), Ndumo (26.9342° S, 32.2824° E), Emkhandlwini (28.508° E, 31.7002° E), Nquthu (28.2195° S, 30.6746° E) and Dundee (28.1650° S, 30.2343° E). The field experiment was conducted over two summer seasons: September 2020–January 2021 and September 2021–January 2022. The experiment was conducted in the vegetable field unit of the Department of Botany, Faculty of Science, Agriculture and Engineering, University of Zululand, KwaDlangezwa campus (28.51° S, 31.50° E) with a sub-tropical climate [56]. The KwaDlangezwa area has a daily mean temperature of 28.4 °C in summer and 14.5 °C in winter [57]. The study area receives an annual rainfall ranging from 299.95 to 350.02 mm [58].

The experiment adopted the randomized block design generated by R 4.2.1 software in the RStudio platform [59]. Seeds were directly sown onto a 10 cm deep pit with fertilizer NPK 2:3:4(30) applied at planting at a rate of 400 kg/ha (40 g/m^2^ per pit) and placed below the seeds in 10–15 cm deep pits. Experimental plots were 3 m × 4 m in size, and seeds were spaced with an intra-row spacing of 1 m and an inter-row spacing of 2 m. Each plot had 20 plants with a net plot of 6 m^2^ having 6 plants. Each of the 18 landraces had 3 replicate plots, which resulted in 54 plots in total, bearing a total of 1080 plants. Weeding and insecticide applications were performed when necessary. The field was irrigated to field capacity for the duration of the experiment using a sprinkler system.

### 4.2. Shoot Traits

Vegetative shoot tips were harvested when landraces were at a fourteen-true-leaf stage. Ten shoots were harvested for each plot per harvest. The shoot tip harvesting was conducted over 28 days at 7-day intervals, resulting in four data sets (42, 49, 56 and 63 days after sowing). Shoot tips were pruned at the third mature leaf from the apical bud. The harvested shoot tips of each harvesting period were used to determine the length, width, fresh and dry mass, as well as moisture content. 

The lengths (mm) of harvested shoot tips per plot were determined using a plastic measuring tape. The fresh mass (g) of the harvested shoots of each plot was determined using the Kern 3 kg analytical scale EWJ 3000-2. The harvested shoots were dried in an oven (Labcon incubator, Model 5016LC) at 65 °C until the constant dry mass was obtained. Thereafter, the shoot moisture content (%) was determined using the formula described by [60]:(1)$$\mathrm{Shoot}\;\mathrm{moisture}\;\mathrm{content}\;(\%)=\frac{Fresh\;mass-dry\;mass}{Freshmass}\times 100$$

### 4.3. Peduncle and Fruit Traits 

Plants in the net plots were used to measure the peduncle and fruit traits of each landrace. Fruits were identified at anthesis (0 days after anthesis (0 DAA) and had their peduncle length (mm) measured from the leaf axil to the base of the inferior ovary using a plastic measuring tape. The peduncle width (mm) was measured using vernier calipers. Similarly, the length (mm) and width (mm) of the same developing fruits were measured using a plastic measuring tape and vernier calipers, respectively. Measurements of the length and width of peduncles and fruits were continued at 1, 2, 3, 4 and 5 DAA while fruits were still attached to the plants.

A total of 5 fruits were also harvested in the net plot of each landrace at 7, 14, 21 and 28 DAA. Harvested fruits had their mass (kg) measured using the analytical weighing scale, as well as their length (mm) and width (mm) measured using vernier calipers. 

### 4.4. Data Analysis

Data were subjected to ANOVA using the GenStat 15th edition, and 10 fruits (*n* = 10) were analyzed for each parameter per landrace. Means were separated using Tukey’s HSD test at a 5% significance level. Correlations and a principal component analysis (PCA) were implemented to determine multi-character variation. The ‘pairs. panels’ function in ‘psych’ R package (North-western University, Evanston, IL, USA) was used for analyzing the correlations within and among shoot, peduncle and fruit traits at different stages of growth (https://cran.r-project.org/web/packages/psych/) (accessed on 27 July 2022) [45]. A cluster analysis with a biplot and dendrogram was conducted to study the variations among landraces using XLSTAT 2022. 

#### Estimation of Variance Components

The phenotypic, genotypic and environmental variances and coefficient of variation were calculated according to the formula described by [61] and cited by [18], as follows:Environmental variance (δ^2^e) = MSE
$$\mathrm{Genotypic}\;\mathrm{variance}\;(\delta^{2}g)=-\frac{MSG-MSE}{r}$$
Phenotypic variance (δ^2^p) = δ^2^g + δ^2^e
where MSG is the mean square due to genotype; MSE is the mean square of error (environmental variance); and (r) is the number of replications.
$$\text{Phenotypic}\;\text{coefficient}\;\text{of}\;\text{variation}\;(\text{PCV})=\surd \frac{\delta^{2}p}{x}\times 100$$
$$\text{Genotypic}\;\text{coefficient}\;\text{of}\;\text{variation}\;(\text{GCV})=\surd \frac{\delta^{2}g}{x}\times 100$$
where

δ^2^p = phenotypic variation

δ^2^g = genotypic variation

x = grand mean of the character studied.

The estimation of heritability in the broad sense was conducted as follows: broad sense heritability (H^2^), expressed as the percentage of the ratio of the genotypic variance (δ^2^g) to the phenotypic variance (δ^2^p), according to [62], was calculated with the following formula:$$h{\;}^{2}=\frac{\delta^{2}g}{\delta^{2}p}\times 100$$

Heritability values estimated higher than 50% were deemed significant estimates [18].

Genetic advance (GA) was estimated as per the formula provided by [62] and cited by [63]:$$\mathrm{GA}=k\;\times \;\surd \frac{\delta^{2}p\times \delta^{2}g}{\delta^{2}p}$$
where

GA = expected genetic advance

δ^2^p = phenotypic variation

δ^2^g = genotypic variation

k = the standard selection differential at 5% selection intensity (k = 2.063).

## 5. Conclusions

Almost all landraces had harvestable shoots at 42 DAS, which could also be harvested at seven-day intervals, with few exceptions. Landraces BG-24, BG-27, BG-31, BG-100/GC, ESC, KRI and NRC had vigorous shoot growth, whereas DSI, KSC, KSP and NSRP had the largest and heaviest fruits. Peduncles became shorter and wider while the fruits elongated and widened from 0 to 5 DAA in all landraces. Shoot traits correlated positively with each other at all stages of growth. However, peduncle length had a negative correlation with peduncle width both in the correlation matrix and principal component analysis. Shoot traits were the main contributors to variability among landraces followed by fruit traits, based on the principal component analysis. The biplot and dendrogram grouped landraces according to shoot traits, fruit traits, as well as shoot regenerative rate and fruit abortion rate. The phenotypic variation in most traits depended on genetic differences. However, traits that were chiefly affected by environmental variability had higher heritability. All traits had low (<1) genetic advancement. This study recorded for the first time the ideal time to initiate (42–49 DAS) and progressively (7–14-day intervals) harvest shoots from *Lagenaria siceraria* landraces. It is also the first record of the relationship between changes that occur in peduncle and fruit size during growth.

## Figures and Tables

**Figure 1 plants-12-00532-f001:**
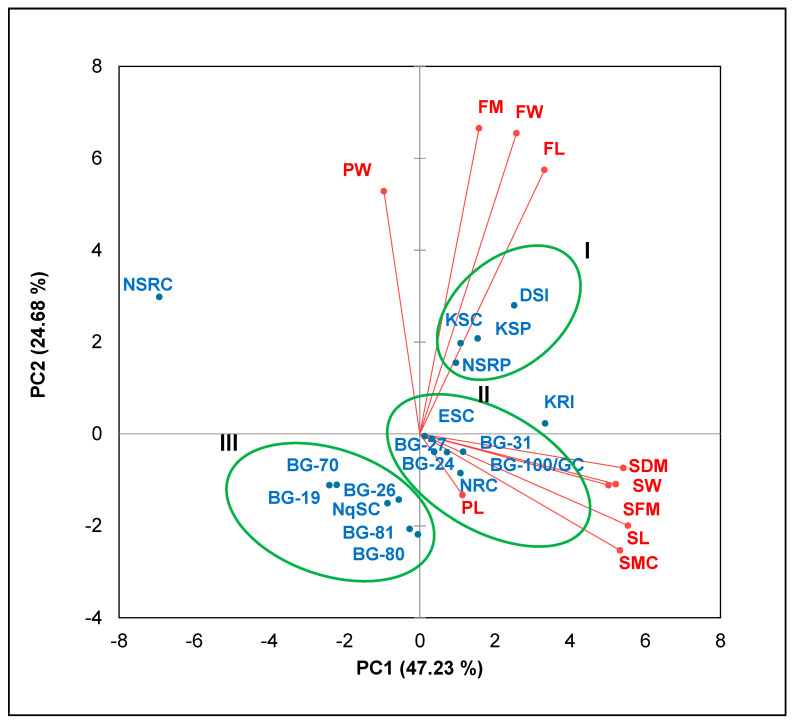
Biplot for fruit and peduncle traits of *L. siceraria* landraces in different growth periods. Landraces are described in Table 1. SL—shoot length; SW—shoot width; SFM—shoot fresh mass; SDM—shoot dry mass; SMC—shoot moisture content; PL—peduncle length; PW—peduncle width; FL—fruit length; FW—fruit width; FM—fruit mass.

**Figure 2 plants-12-00532-f002:**
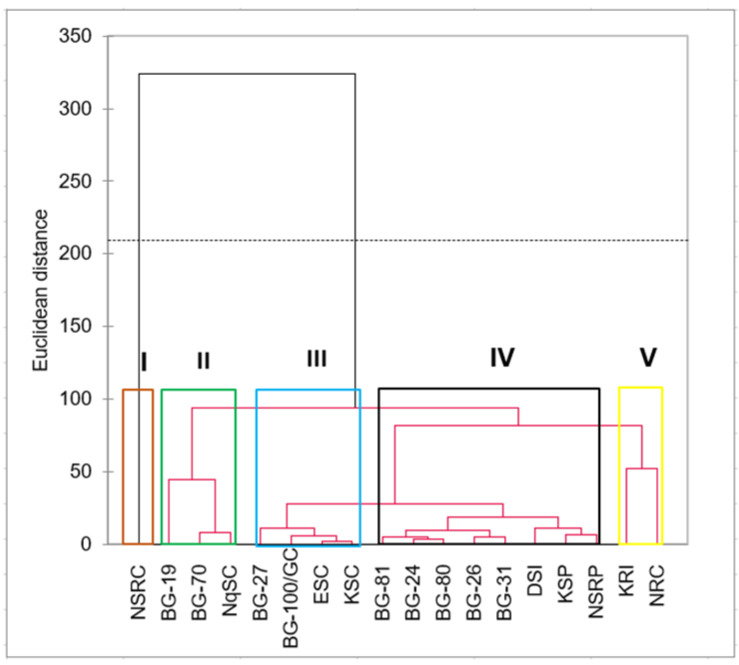
Agglomerative hierarchical cluster showing dissimilarities amongst *L. siceraria* landraces using unweighted pair group method with arithmetic mean method (UPGMA). Landraces are described in Table 1.

**Table 1 plants-12-00532-t001:** Description of landraces from northern KwaZulu-Natal and Limpopo according to their origin as well as fruit and seed morphology.

Prov	LR	Area	Fruit Color	Fruit Texture	Fruit Shape	Seed Type	Seed Color	Seed Texture	Seed Size	Seed Line	Seed Shape
KZN	KSP	Khangelani	Pale green	Smooth	Pear	Asiatica	Brown	Leathery	Large	Present	Slightly oblong to rectangular
KZN	KSC	Khangelani	Pale green	Smooth	Curvilinear	Asiatica	Brown	Leathery	Large	Present	Slightly oblong to rectangular
KZN	KRI	Khangelani	Green	Rough	Isodiametric	Siceraria	Dark brown	Leathery	Large	Present	Slightly oblong to rectangular
KZN	NRC	Ndumo	Dark green	Rough	Cylindrical	Siceraria	Creamy brown	Smooth	Small	Absent	Oblong
KZN	NSRC	Nquthu	Green	Semi-rough	Curvilinear	Intermediate	Brown	Leathery	Medium	Present	Slightly oblong
KZN	NSRP	Nquthu	Pale green	Semi-rough	Pear	Intermediate	Brown	Leathery	Medium	Present	Slightly oblong
KZN	NqSC	Nquthu	Pale green	Smooth	Semi-curvilinear	Asiatica	Light brown	Leathery	Medium	Present	Slightly oblong
KZN	DSI	Dundee	Dark green	Smooth	Isodiametric	Siceraria	Dark brown	Smooth	Large	Present	Oblong
KZN	ESC	Emkhandlwini	Pale green	Smooth	Curvilinear	Asiatica	Light brown	Leathery	Medium	Present	Slightly oblong
LP	BG-19	Kgohloane	Dark green	Smooth	Isodiametric	Siceraria	Brown	Leathery	Large	Present	Slightly oblong to rectangular
LP	BG-24	Go-Phasa	Pale green	Corrugate	Cavate	Siceraria	Dark brown	Smooth	Small	Absent	Oblong
LP	BG-26	Kgohloane	Dark green	Smooth	Cavate	Intermediate	Brown	Leathery	Large	Present	Slightly oblong to rectangular
LP	BG-27	Kgohloane	Pale and dark green	Semi-rough	Cavate	Siceraria	Brown	Leathery	Small	Present	Oblong
LP	BG-31	Kgohloane	Dark green	Smooth	Cavate	Intermediate	Brown	Leathery	Large	Present	Slightly oblong to rectangular
LP	BG-70	Go-Phasa	Pale green	Smooth	Pyriform	Asiatica	Light brown	Leathery	Large	Present	Slightly oblong to rectangular
LP	BG-80	Moletjie-Mabokelele	Pale green	Corrugate	Cavate	Asiatica	Dark brown	Smooth	Medium	Absent	Oblong
LP	BG-81	Kgohloane	Pale and dark green	Corrugate	Cavate	Asiatica	Brown	Leathery	Large	Present	Oblong
LP	BG-100/GC	Kgohloane	Pale green	Semi-rough	Cylindrical	Asiatica	Light brown	Leathery	Medium	Present	Slightly oblong

Prov—province, KZN—KwaZulu-Natal and LP—Limpopo. LR—landraces. BG—bottle gourd. The area of origin, fruit texture and fruit shape were used to name the landraces from KZN. The name of each landrace from Limpopo was coined using the entry number based on the area of origin [3,22,23,24].

**Table 2 plants-12-00532-t002:** Traits of *Lagenaria siceraria* shoots harvested at different stages of growth (*n* = 10).

Harvest Period (Days after Sowing)					
Traits	Landraces	42 DAS	49 DAS	56 DAS	63 DAS
SL	BG-19	NA	232.0 ^lmn^	299.0 ^d–n^	300.0 ^d–n^
	BG-24	263.5 ^h–n^	307.0 ^d–n^	347.5 ^b–m^	349.0 ^b–l^
	BG-26	252.0 ^j–n^	337.5 ^b–n^	347.0 ^b–m^	377.0 ^a–h^
	BG-27	236.8 ^k–n^	286.4 ^d–n^	376.1 ^a–h^	333.0 ^b–n^
	BG-31	238.5 ^k–n^	327.5 ^c–n^	341.2 ^b–n^	371.5 ^a–j^
	BG-70	276.5 ^e–n^	NA	332.5 ^b–n^	391.0 ^a–f^
	BG-80	278.0 ^e–n^	328.5 ^c–n^	306.5 ^d–n^	335.5 ^b–n^
	BG-81	252.2 ^j–n^	254.3 ^i–n^	376.0 ^a–h^	396.0 ^a–e^
	BG-100/GC	228.2 ^mn^	264.4 ^g–n^	328.5 ^c–n^	365.0 ^a–j^
	DSI	404.2 ^a–d^	367.0 ^a–j^	313.0 ^c–n^	296.5 ^d–n^
	ESC	226.9 ^n^	319.9 ^c–n^	289.9 ^d–n^	367.5 ^a–j^
	KRI	375.0 ^a–h^	400.0 ^a–d^	450.0 ^ab^	476.0 ^a^
	KSC	373.9 ^a–i^	223.0 ^n^	293.0 ^d–n^	287.9 ^d–n^
	KSP	430.4 ^abc^	297.0 ^d–n^	278.5 ^e–n^	327.0 ^c–n^
	NqSC	370.5 ^a–j^	NA	299.0 ^d–n^	330.0 ^c–n^
	NRC	273.6 ^f–n^	382.0 ^a–h^	378.0 ^a–h^	406.0 ^a–d^
	NSRP	268.5 ^g–n^	356.0 ^b–k^	342.5 ^b–n^	384.0 ^a–g^
SW	BG-19	NA	7.0 ^de^	6.9 ^de^	5.9 ^e^
	BG-24	4.9 ^e^	6.2 ^e^	6.1 ^e^	7.1 ^de^
	BG-26	7.9 ^cde^	6.9 ^de^	7.0 ^de^	7.1 ^de^
	BG-27	5.5 ^e^	6.2 ^e^	6.1 ^e^	6.9 ^de^
	BG-31	5.6 ^e^	7.0 ^de^	14.3 ^ab^	7.2 ^de^
	BG-70	5.0 ^e^	NA	6.5 ^e^	7.0 ^de^
	BG-80	6.6 ^de^	5.9 ^e^	7.6 ^de^	6.6 ^de^
	BG-81	6.7 ^de^	7.0 ^de^	6.7 ^de^	7.5 ^de^
	BG-100/GC	6.8 ^de^	7.4 ^de^	8.3 ^cde^	8.6 ^cde^
	DSI	6.7 ^de^	7.0 ^de^	6.7 ^de^	6.6 ^de^
	ESC	6.5 ^e^	6.4 ^e^	6.2 ^e^	6.2 ^e^
	KRI	6.8 ^de^	13.3 ^abc^	11.9 ^bcd^	10.3 ^b–e^
	KSC	6.6 ^de^	6.2 ^e^	6.8 ^de^	7.8 ^de^
	KSP	5.9 ^e^	17.6 ^a^	6.8 ^de^	7.0 ^de^
	NqSC	4.9 ^e^	NA	7.0 ^de^	9.0 ^b–e^
	NRC	4.9 ^e^	5.7 ^e^	7.1 ^de^	7.3 ^de^
	NSRP	5.5 ^e^	7.0 ^de^	6.8 ^de^	7.1 ^de^
SFM	BG-19	NA	15.23 ^g–o^	15.63 ^g–o^	15.48 ^g–o^
	BG-24	16.73 ^f–o^	17.14 ^e–o^	13.51 ^i–o^	13.85 ^i–o^
	BG-26	6.29 ^o^	12.69 ^k–o^	14.42 ^h–o^	18.98 ^d–m^
	BG-27	7.71 ^no^	10.69 ^l–o^	24.06 ^a–i^	13.00 ^j–o^
	BG-31	9.29 ^m–o^	18.18 ^d–n^	12.79 ^k–o^	17.44 ^e–n^
	BG-70	9.08 ^m–o^	NA	12.98 ^j–o^	18.78 ^d–m^
	BG-80	11.99 ^k–o^	14.85 ^g–o^	14.40 ^h–o^	16.84 ^f–o^
	BG-81	10.59 ^l–o^	14.07 ^i–o^	17.58 ^e–n^	19.61 ^d–m^
	BG-100/GC	13.31 ^i–o^	16.61 ^f–o^	17.30 ^e–n^	26.82 ^a–f^
	DSI	32.68 ^ab^	25.33 ^a–h^	15.71 ^g–o^	17.57 ^e–n^
	ESC	10.44 ^l–o^	15.12 ^g–o^	11.41 ^l–o^	15.97 ^f–o^
	KRI	29.00 ^a–d^	25.50 ^a–g^	34.75 ^a^	30.62 ^abc^
	KSC	15.39 ^g–o^	13.27 ^i–o^	13.09 ^i–o^	16.73 ^f–o^
	KSP	11.43 ^l–o^	13.99 ^i–o^	12.66 ^k–o^	15.88 ^f–o^
	NqSC	28.05 ^a–e^	NA	13.79 ^i–o^	22.73 ^b–k^
	NRC	31.79 ^abc^	23.80 ^a–j^	21.17 ^c–l^	26.87 ^a–f^
	NSRP	9.81 ^m–o^	16.84 ^f–o^	13.85 ^i–o^	22.64 ^b–k^
SDM	BG-19	NA	1.87 ^d–n^	2.17 ^c–l^	1.30 ^j–n^
	BG-24	2.19 ^c–k^	1.99 ^c–m^	1.68 ^e–n^	1.30 ^j–n^
	BG-26	1.54 ^g–n^	1.68 ^e–n^	1.77 ^d–m^	1.58 ^g–n^
	BG-27	1.44 ^h–n^	1.27 ^k–n^	1.58 ^g–n^	0.98^n^
	BG-31	1.43 ^h–n^	2.02 ^c–m^	1.66 ^e–n^	1.37 ^j–n^
	BG-70	1.68 ^e–n^	NA	1.59 ^g–n^	1.67 ^e–n^
	BG-80	1.81 ^d–m^	1.95 ^c–m^	1.38 ^i–n^	1.55 ^g–n^
	BG-81	1.97 ^c–m^	1.73 ^e–n^	2.02 ^c–m^	1.49 ^g–n^
	BG-100/GC	2.29 ^b–i^	2.11 ^c–l^	2.60 ^a–e^	2.13 ^c–l^
	DSI	3.23 ^ab^	2.54 ^b–f^	2.21 ^c–j^	1.87 ^d–n^
	ESC	1.85 ^d–n^	2.02 ^c–m^	1.25 ^l–n^	1.47 ^h–n^
	KRI	2.32 ^b–h^	2.41 ^b–g^	3.50^a^	2.54 ^b–f^
	KSC	2.69 ^a–d^	1.70 ^e–n^	1.82 ^d–m^	1.67 ^e–n^
	KSP	2.56 ^b–f^	1.76 ^d–n^	2.08 ^c–l^	1.86 ^d–n^
	NqSC	2.85 ^abc^	NA	1.73 ^e–n^	1.89 ^d–n^
	NRC	2.31 ^b–i^	2.57 ^a–f^	2.08 ^c–l^	2.08 ^c–l^
	NSRP	1.48 ^g–n^	2.09 ^c–l^	1.65 ^f–n^	1.12^mn^
SMC	BG-19	NA	87.70 ^d–m^	85.80 ^j–r^	85.94 ^ab^
	BG-24	87.00 ^g–p^	88.60 ^b–l^	87.00 ^g–p^	87.36 ^ab^
	BG-26	74.20 ^u^	85.90 ^j–r^	86.90 ^g–p^	85.80 ^ab^
	BG-27	80.20 ^st^	88.10 ^c–l^	89.60 ^a–k^	90.27 ^ab^
	BG-31	84.30 ^l–s^	89.00 ^b–l^	85.90 ^j–r^	86.78 ^ab^
	BG-70	81.60 ^qrs^	NA	87.30 ^e–n^	87.48 ^ab^
	BG-80	85.00 ^k–s^	86.90 ^g–p^	90.30 ^a–j^	90.42 ^a^
	BG-81	81.20 ^rst^	87.60 ^d–m^	87.90 ^c–l^	87.85 ^ab^
	BG-100/GC	82.60 ^n–s^	87.00 ^g–p^	84.40 ^l–s^	84.94 ^ab^
	DSI	89.40 ^a–k^	89.50 ^a–k^	86.20 ^i–q^	86.24 ^ab^
	ESC	82.30 ^p–s^	86.80 ^h–p^	88.40 ^b–l^	87.88 ^ab^
	KRI	92.10 ^a–e^	90.20 ^a–j^	89.70 ^a–k^	89.83 ^ab^
	KSC	82.40 ^o–s^	87.00 ^g–p^	85.80 ^j–r^	85.62 ^ab^
	KSP	76.60 ^tu^	87.40 ^e–n^	83.00 ^m–s^	83.48 ^b^
	NqSC	90.00 ^a–j^	NA	87.20 ^f–o^	87.25 ^ab^
	NRC	93.00 ^ab^	89.40 ^a–k^	90.10 ^a–j^	89.88 ^ab^
	NSRP	84.90 ^k–s^	87.80 ^d–m^	88.10 ^c–l^	88.14 ^ab^

Mean values with different letter(s) within columns and rows of each trait differ significantly at *p* < 0.05 according to Tukey’s Honest Significant Difference test. Missing data for cases where shoots were not harvestable are denoted as “NA” for “not available”. Landraces are described in Table 1. SL—shoot length (mm); SW—shoot width (mm); SFM—shoot fresh mass (g); SDM—shoot dry mass (g); SMC—shoot moisture content (%).

**Table 3 plants-12-00532-t003:** Peduncle and fruit traits of intact *Lagenaria siceraria* fruits (*n* = 10).

Traits	Landraces		Days after Anthesis				
		0	1	2	3	4	5
PL	BG-19	36.3 ^o–B^	29.9 ^r–B^	27.8 ^s–B^	25.1 ^v–B^	20.4 ^z–B^	17.1 ^B^
	BG-24	85.4 ^bcd^	75.0 ^b–f^	67.5 ^c–k^	66.8 ^c–k^	63.3 ^d–m^	54.4 ^e–r^
	BG-26	46.6 ^g–x^	35.4 ^o–P^	30.0 ^r–B^	26.4 ^t–B^	27.9 ^s–B^	24.0 ^w–B^
	BG-27	99.8 ^ab^	90.1 ^abc^	72.9 ^c–g^	55.2 ^e–r^	38.0 ^m–B^	45.6 ^i–A^
	BG-31	67.3 ^c–k^	52.3 ^f–t^	42.9 ^j–B^	57.8 ^e–p^	37.0 ^n–B^	25.8 ^u–B^
	BG-70	36.8 ^n–B^	30.0 ^r–B^	27.0 ^s–B^	23.8 ^w–B^	22.5 ^x–B^	20.0 ^AB^
	BG-80	57.1 ^e–p^	50.2 ^f–v^	45.1 ^i–A^	43.6 ^j–A^	39.3 ^l–B^	36.3 ^o–B^
	BG-81	72.5 ^c–h^	54.6 ^f–v^	49.5 ^f–w^	46.6 ^h–y^	46.2 ^i–z^	42.5 ^j–B^
	BG-100/GC	111.5 ^a^	79.4 ^b–e^	65.5 ^c–k^	57.8 ^e–p^	52.7 ^f–s^	51.6 ^f–u^
	DSI	68.1 ^c–j^	51.2 ^f–u^	39.5 ^l–B^	30.9 ^q–B^	21.1 ^y–B^	20.7 ^y–B^
	ESC	72.7 ^c–g^	56.3 ^e–q^	48.4 ^g–x^	48.5 ^g–w^	43.1 ^j–A^	37.7 ^m–B^
	KRI	64.4 ^c–l^	62.3 ^d–n^	57.8 ^e–p^	34.3 ^o–B^	30.9 ^o–B^	27.3 ^s–B^
	KSC	70.8 ^c–i^	58.7 ^e–o^	54.4 ^e–r^	48.3 ^g–x^	43.5 ^j–A^	36.8 ^n–B^
	KSP	56.4 ^e–q^	41.9 ^k–B^	35.2 ^o–B^	32.1 ^p–B^	30.5 ^r–B^	30.0 ^r–B^
	NqSC	NA	NA	NA	NA	NA	NA
	NRC	NA	NA	NA	NA	NA	NA
	NSRC	NA	NA	NA	NA	NA	NA
	NSRP	48.2 ^g–x^	41.8 ^k–B^	36.5 ^n–B^	33.7 ^o–B^	27.8 ^s–B^	24.0 ^w–B^
PW	BG-19	3.8 ^r–B^	4.3 ^m–B^	4.6 ^i–A^	5.1 ^d–u^	5.6 ^c–o^	7.3^a^
	BG-24	3.6 ^v–B^	3.8 ^r–B^	4.3 ^k–B^	4.9 ^e–x^	4.7 ^g–z^	5.3 ^c–q^
	BG-26	3.7 ^u–B^	4.2 ^o–B^	4.7 ^h–A^	5.3 ^c–q^	5.7 ^b–l^	7.2^a^
	BG-27	3.2^AB^	3.6 ^v–B^	4.2 ^o–B^	4.4 ^j–B^	5.0 ^d–v^	5.3 ^c–p^
	BG-31	3.8 ^r–B^	4.1 ^o–B^	4.6 ^i–A^	5.3 ^c–p^	5.5 ^c–p^	5.6 ^c–o^
	BG-70	3.4 ^y–B^	3.5 ^x–B^	4.5 ^i–B^	5.2 ^c–s^	5.2 ^c–s^	6.1 ^a–h^
	BG-80	3.1 ^B^	3.6 ^w–B^	4.1 ^p–B^	4.4 ^k–B^	4.8 ^f–y^	4.9 ^d–x^
	BG-81	3.3 ^zAB^	3.8 ^s–B^	4.2 ^o–B^	4.4 ^j–B^	4.5 ^i–B^	5.1 ^c–t^
	BG-100/GC	3.7 ^t–B^	4.2 ^o–B^	4.6 ^i–A^	4.9 ^d–x^	5.5 ^c–p^	5.7 ^b–m^
	DSI	3.7 ^t–B^	4.4 ^k–B^	5.0 ^d–w^	5.7 ^b–n^	5.8 ^a–j^	5.9 ^a–i^
	ESC	3.4 ^y–B^	3.9 ^q–B^	4.5 ^i–B^	4.4 ^j–B^	5.9 ^a–i^	5.1 ^d–u^
	KRI	4.5 ^i–B^	4.9 ^d–x^	5.2 ^c–r^	5.9 ^a–i^	6.2 ^a–f^	6.6 ^abc^
	KSC	3.6 ^x–B^	4.3 ^l–B^	4.7 ^g–z^	5.7 ^b–m^	6.1 ^a–g^	6.1 ^a–g^
	KSP	4.1 ^o–B^	4.9 ^f–x^	5.7 ^b–k^	5.7 ^b–m^	6.3 ^a–e^	7.0 ^ab^
	NqSC	NA	NA	NA	NA	NA	NA
	NRC	NA	NA	NA	NA	NA	NA
	NSRC	NA	NA	NA	NA	NA	NA
	NSRP	3.8 ^r–B^	4.2 ^o–B^	4.8 ^f–y^	5.4 ^c–p^	5.7 ^b–l^	6.3 ^a–d^
FL	BG-19	15.9 ^t^	17.0 ^st^	19.7 ^o–t^	22.1 ^n–t^	27.6 ^i–t^	34.1 ^e–t^
	BG-24	26.0 ^k–t^	34.2 ^e–t^	45.2 ^d–p^	41.0 ^d–t^	42.0 ^d–t^	56.4 ^c–f^
	BG-26	17.6 ^r–t^	21.1 ^o–t^	24.0 ^l–u^	27.2 ^j–t^	34.7 ^e–t^	42.7 ^d–s^
	BG-27	25.9 ^k–t^	40.7 ^d–t^	49.7 ^c–l^	39.2 ^e–t^	59.3 ^cde^	102.5 ^a^
	BG-31	19.0 ^p–t^	33.4 ^e–t^	37.6 ^e–t^	42.7 ^d–s^	49.3 ^c–m^	53.8 ^c–i^
	BG-70	18.3 ^q–t^	24.1 ^l–t^	27.2 ^j–t^	29.1 ^g–t^	35.0 ^e–t^	43.5 ^d–r^
	BG-80	21.6 ^o–t^	26.3 ^k–t^	29.9 ^g–t^	30.5 ^f–t^	33.2 ^e–t^	41.6 ^d–t^
	BG-81	21.1 ^o–t^	24.2 ^l–t^	28.8 ^h–t^	28.4 ^h–t^	34.9 ^e–t^	40.5 ^d–t^
	BG-100/GC	18.8 ^q–t^	21.4 ^o–t^	23.3 ^m–t^	27.0 ^j–t^	28.7 ^h–t^	31.6 ^f–t^
	DSI	31.6 ^f–t^	36.1 ^e–t^	39.9 ^d–t^	43.8 ^d–r^	51.9 ^c–k^	74.5^bc^
	ESC	33.3 ^e–t^	42.0 ^d–t^	51.4 ^c–k^	53.0 ^c–j^	44.6 ^d–q^	65.8^cd^
	KRI	21.5 ^o–t^	24.1 ^l–t^	24.5 ^l–u^	26.8 ^j–t^	31.1 ^f–t^	33.3 ^e–t^
	KSC	27.9 ^i–t^	33.1 ^e–t^	33.8 ^e–t^	50.0 ^c–l^	48.4 ^c–n^	54.6 ^c–h^
	KSP	25.9 ^k–t^	45.3 ^d–p^	45.4 ^d–o^	49.7 ^c–l^	58.3^cde^	55.2 ^c–g^
	NqSC	NA	NA	NA	NA	NA	NA
	NRC	NA	NA	NA	NA	NA	NA
	NSRC	NA	NA	NA	NA	NA	NA
	NSRP	35.4 ^e–t^	35.4 ^e–t^	41.2 ^d–t^	45.5 ^d–o^	56.5 ^c–f^	95.2^ab^
FW	BG-19	15.2 ^n–t^	15.9 ^m–t^	21.2 ^h–t^	23.0 ^g–t^	28.5 ^e–o^	32.7 ^c–i^
	BG-24	15.1 ^n–t^	18.9 ^h–t^	27.7 ^e–r^	22.6 ^g–t^	22.7 ^g–t^	31.1 ^c–k^
	BG-26	13.4 ^p–t^	18.2 ^i–t^	17.9 ^i–t^	20.8 ^h–t^	29.4 ^d–n^	36.9 ^c–g^
	BG-27	13.0 ^q–t^	17.8 ^i–t^	26.3 ^e–s^	18.1 ^i–t^	25.5 ^f–t^	43.6^bcd^
	BG-31	11.1 ^t^	13.4 ^p–t^	17.3 ^j–t^	29.2 ^d–n^	40.8 ^b–e^	54.7^ab^
	BG-70	13.4 ^p–t^	16.0 ^l–t^	18.3 ^i–t^	18.0 ^i–t^	21.2 ^h–t^	24.2 ^g–t^
	BG-80	14.0 ^o–t^	16.0 ^m–t^	18.2 ^i–t^	19.6 ^h–t^	23.9 ^g–t^	27.8 ^e–r^
	BG-81	13.2 ^q–t^	15.5 ^n–t^	21.8 ^h–t^	22.1 ^g–t^	24.8 ^g–t^	31.5 ^c–j^
	BG-100/GC	13.8 ^o–t^	15.7 ^n–t^	17.1 ^j–t^	19.4 ^h–t^	21.6 ^h–t^	24.2 ^g–t^
	DSI	16.5 ^k–t^	24.5 ^g–t^	28.2 ^e–p^	33.9 ^c–h^	44.0^bcd^	58.7^a^
	ESC	10.8 ^tu^	13.8 ^o–t^	18.0 ^i–t^	22.2 ^g–t^	15.3 ^n–t^	24.4 ^g–t^
	KRI	17.3 ^j–t^	19.9 ^h–t^	20.9 ^h–t^	22.6 ^g–t^	25.1 ^f–t^	30.8 ^d–m^
	KSC	15.2 ^n–t^	21.7 ^h–t^	19.5 ^h–t^	23.9 ^g–t^	28.0 ^e–q^	40.0 ^b–f^
	KSP	11.4 ^st^	15.7 ^n–t^	27.5 ^e–r^	31.0 ^d–l^	14.4 ^n–t^	16.8 ^j–t^
	NqSC	NA	NA	NA	NA	NA	NA
	NRC	NA	NA	NA	NA	NA	NA
	NSRC	NA	NA	NA	NA	NA	NA
	NSRP	11.5 ^st^	13.0 ^rst^	14.0 ^o–t^	15.1 ^n–t^	17.1 ^j–t^	46.1^abc^

Mean values with different letter(s) within columns and rows of each trait differ significantly at *p* < 0.05 according to Tukey’s Honest Significant Difference test. Missing data for cases where fruits had high abortion rates and were not measured at early stages of growth are denoted as “NA” for “not available”. Landraces are described in Table 1. PL—peduncle length (mm); PW—peduncle width (mm); FL—fruit length (mm); FW—fruit width (mm).

**Table 4 plants-12-00532-t004:** Traits of sequentially harvested *Lagenaria siceraria* fruits (*n* = 10).

Traits	Landraces		Days after Anthesis		
		7	14	21	28
PL	BG-24	78.67 ^a–e^	72.33 ^a–h^	69.33 ^b–j^	61.00 ^b–m^
	BG-27	42.83 ^f–n^	63.50 ^b–l^	75.83 ^a–g^	66.00 ^b–k^
	BG-31	43.60 ^f–n^	47.67 ^d–n^	31.00^lmn^	27.17^mn^
	BG-100/GC	65.17 ^b–l^	63.00 ^b–l^	72.50 ^a–h^	62.55 ^b–l^
	DSI	81.17 ^a–d^	57.00 ^c–n^	88.00^abc^	75.50 ^a–g^
	ESC	45.25 ^e–n^	74.00 ^a–g^	46.92 ^d–n^	49.67 ^d–n^
	KRI	61.83 ^b–m^	51.67 ^d–n^	44.67 ^e–n^	26.17^n^
	KSC	69.83 ^a–j^	57.67 ^c–n^	67.00 ^b–j^	75.32 ^a–g^
	KSP	66.83 ^b–j^	104.00 ^a^	93.67 ^ab^	86.83 ^abc^
	NqSC	78.67 ^a–e^	66.33 ^b–j^	62.50 ^b–l^	55.00 ^c–n^
	NRC	92.50^ab^	76.50 ^a–f^	73.50 ^a–h^	70.67 ^a–i^
	NSRP	35.67 ^j–n^	31.67 ^k–n^	41.22 ^g–n^	43.00^f—n^
	NSRC	39.33 ^h–n^	72.67 ^a–h^	78.67 ^a–e^	36.50 ^i–n^
PW	BG-24	3.8^tu^	4.7 ^q–u^	5.0 ^o–u^	7.2 ^d–k^
	BG-27	4.8 ^p–u^	3.6 ^u^	4.1 ^stu^	5.5 ^l–s^
	BG-31	4.3 ^r–u^	5.0 ^o–u^	5.8 ^k–r^	7.0 ^e–l^
	BG-100/GC	5.1 ^o–u^	5.3 ^m–s^	6.3 ^i–p^	8.7 ^abc^
	DSI	6.0 ^k–q^	8.0 ^a–h^	7.8 ^b–h^	8.7 ^abc^
	ESC	4.1^stu^	4.0^stu^	4.2 ^r–u^	5.0 ^o–u^
	KRI	7.1 ^e–k^	8.2 ^a–g^	8.5 ^a–e^	9.5 ^a^
	KSC	7.0 ^e–k^	8.0 ^a–h^	7.0 ^e–l^	7.0 ^e–k^
	KSP	6.0 ^k–q^	5.3 ^n–t^	7.1 ^e–k^	6.8 ^f–m^
	NqSC	3.8 ^tu^	6.7 ^g–n^	7.7 ^b–j^	9.0 ^abc^
	NRC	8.4 ^a–e^	8.3 ^a–f^	9.2 ^ab^	8.7 ^abc^
	NSRP	6.0 ^k–q^	6.5 ^h–o^	6.8 ^f–m^	7.0 ^e–l^
	NSRC	6.2 ^j–p^	7.6 ^c–j^	7.9 ^b–h^	7.0 ^e–l^
FL	BG-24	88.3 ^s–v^	128.7 ^o–t^	127.7 ^o–t^	138.0 ^n–s^
	BG-27	108.3 ^r–v^	127.8 ^o–t^	123.0 ^p–u^	220.5 ^f–l^
	BG-31	119.7 ^p–u^	156.1 ^m–r^	207.0 ^g–m^	213.3 ^g–m^
	BG-100/GC	64.5 ^uv^	84.7 ^s–v^	108.3 ^r–v^	218.3 ^g–l^
	DSI	57.5 ^v^	177.0 ^k–p^	239.4 ^e–j^	256.7 ^e–h^
	ESC	96.7 ^s–v^	130.9 ^o–s^	227.9 ^e–k^	237.7 ^e–j^
	KRI	65.2 ^uv^	125.7 ^o–t^	192.0 ^i–n^	220.0 ^f–l^
	KSC	134.8 ^n–s^	286.0^de^	351.0 ^bc^	374.6 ^bc^
	KSP	158.8 ^m–r^	161.8 ^l–r^	320.5 ^cd^	389.7 ^b^
	NqSC	88.3 ^s–v^	172.5 ^k–q^	200.2 ^h–m^	261.0 ^efg^
	NRC	69.7^tuv^	113.3 ^q–v^	132.3 ^o–s^	226.7 ^f–k^
	NSRP	126.0 ^o–t^	182.3 ^j–o^	277.9 ^def^	597.0^a^
	NSRC	139.3 ^n–s^	204.7 ^g–m^	245.3 ^e–i^	605.5^a^
FW	BG-24	43.00 ^uvw^	83.00 ^m–p^	91.67 ^h–o^	120.00 ^def^
	BG-27	46.00 ^uvw^	73.67 ^n–s^	90.75 ^i–o^	83.00 ^m–p^
	BG-31	40.00 ^vw^	60.67 ^q–v^	79.50 ^m–q^	96.33 ^g–m^
	BG-100/GC	51.67 ^t–w^	73.83 ^n–r^	88.17 ^k–o^	121.42 ^c–f^
	DSI	36.33 ^w^	95.00 ^g–n^	133.00 ^cd^	175.83 ^a^
	ESC	39.92 ^vw^	75.33 ^m–q^	79.04 ^m–q^	96.13 ^g–m^
	KRI	53.00 ^r–w^	90.83 ^i–o^	107.67 ^f–k^	142.50 ^bc^
	KSC	73.00 ^o–t^	89.00 ^k–o^	155.13 ^ab^	112.67 ^d–h^
	KSP	52.17 ^s–w^	72.33 ^o–t^	111.00 ^e–j^	165.17 ^a^
	NqSC	43.00 ^uvw^	89.50 ^j–o^	105.88 ^f–l^	130.50 ^cde^
	NRC	38.83 ^w^	73.83 ^n–r^	91.67 ^h–o^	127.33 ^c–f^
	NSRP	43.50 ^uvw^	62.73 ^p–u^	85.78 ^l–o^	111.50 ^d–i^
	NSRC	49.50 ^uvw^	88.17 ^k–o^	111.50 ^d–i^	114.50 ^d–g^
FM	BG-24	0.057 ^s^	0.270 ^k–s^	0.370 ^j–r^	0.690 ^f–i^
	BG-27	0.073 ^rs^	0.212 ^n–s^	0.354 ^j–s^	0.437 ^i–p^
	BG-31	0.066 ^rs^	0.159 ^o–s^	0.347 ^j–s^	0.523 ^g–m^
	BG-100/GC	0.091 ^qrs^	0.252 ^l–s^	0.462 ^h–o^	1.257 ^cd^
	DSI	0.082 ^qrs^	0.553 ^g–l^	1.332 ^cd^	2.277 ^a^
	ESC	0.053 ^s^	0.231 ^m–s^	0.296 ^k–s^	0.471 ^h–n^
	KRI	0.080 ^qrs^	0.382 ^j–q^	0.788^fg^	1.258 ^cd^
	KSC	0.152 ^p–s^	0.630 ^f–j^	1.145^cde^	1.434 ^bc^
	KSP	0.108 ^qrs^	0.190 ^n–s^	0.897 ^ef^	1.655 ^b^
	NqSC	0.057 ^s^	0.320 ^k–s^	0.762 ^fgh^	1.310 ^cd^
	NRC	0.068 ^rs^	0.322 ^k–s^	0.563 ^g–k^	1.345 ^cd^
	NSRP	0.080 ^qrs^	0.530 ^g–m^	0.642 ^f–j^	1.110 ^de^
	NSRC	0.105 ^qrs^	0.532 ^g–m^	1.133 ^cde^	1.180 ^cde^

Mean values with different letter(s) within columns and rows of each trait differ significantly at *p* < 0.05 according to Tukey’s Honest Significant Difference test. Landraces are described in Table 1. PL—peduncle length (mm); PW—peduncle width (mm); FL—fruit length (mm); FW—fruit width (mm); FM, fruit mass (kg).

**Table 5 plants-12-00532-t005:** Correlation coefficients for shoot (**a**), peduncle and fruit (**b**) traits at different growth stages, and combined traits (**c**) of *L. siceraria* landraces (*n* = 10).

**(a)**									
**Traits**		**42 DAS**		**49 DAS**		**56 DAS**		**63 DAS**	
SL × SW		0.43 ***		0.57 ***		0.21 *		0.40 ***	
SL × SFM		0.62 ***		0.83 ***		0.41 ***		0.63 ***	
SL × SDM		0.73 ***		0.82 ***		0.48 ***		0.40 ***	
SL × SMC		0.63 ***		0.82 ***		0.32 ***		0.21 *	
SW × SFM		0.22 **		0.51 ***		ns		0.60 ***	
SW × SDM		0.45 ***		0.52 ***		ns		0.42 ***	
SW × SMC		0.74 ***		0.63 ***		0.21 *		0.19 *	
SFM × SDM		0.79 ***		0.93 ***		0.43 ***		0.68 ***	
SFM × SMC		0.52 ***		0.72 ***		0.41 ***		0.32 **	
SDM × SMC		0.58 ***		0.77 ***		−0.33 **		−0.42 ***	
**(b)**									
**Traits**		**0 DAA**	**1 DAA**	**2 DAA**		**3 DAA**	**4 DAA**	**5 DAA**	
PL × PW		Ns	ns	−0.18 *		−0.22 **	−0.20 *	−0.35 ***	
PL × FL		Ns	ns	ns		ns	ns	ns	
PL × FW		Ns	ns	ns		ns	−0.19 *	−0.17 *	
PW × FL		Ns	ns	ns		ns	0.26 **	ns	
PW × FW		0.27 ***	ns	ns		0.17 *	ns	Ns	
FL × FW		Ns	0.23 **	0.48 ***		0.60 ***	ns	0.60 ***	
**(c)**									
**Variables**	**SL**	**SW**	**SFM**	**SDM**	**SMC**	**PL**	**PW**	**FL**	**FW**
SW	**0.878**								
SFM	**0.845**	**0.695**							
SDM	**0.875**	**0.813**	**0.907**						
SMC	**0.949**	**0.828**	**0.725**	**0.807**					
PL	0.070	0.093	0.133	0.091	0.171				
PW	−0.239	−0.098	−0.045	−0.062	−0.416	−0.582			
FL	0.351	0.392	0.215	0.333	0.377	0.092	0.124		
FW	0.207	0.271	0.108	0.208	0.221	0.072	0.225	**0.984**	
FM	0.003	0.035	0.269	0.266	−0.105	0.151	0.460	**0.605**	**0.662**

DAS—days after sowing; DAA—days after anthesis; SL—shoot length (mm); SW—shoot width (mm); SFM—shoot fresh mass (g); SDM—shoot dry mass (g); SMC—shoot moisture content (%); PL—peduncle length (mm); PW—peduncle width (mm); FL—fruit length (mm); FW—fruit width (mm); FM—fruit mass (kg). In Table 4a,b: ns, not significant; * *p* < 0.05; ** *p* < 0.01; *** *p* < 0.001. In Table 4c: values ˃ 0.6 (in bold) are significant.

**Table 6 plants-12-00532-t006:** Loadings of the variables for the first three principal components.

Variables	PC1	PC2	PC3
SL	**0.943**	−0.245	−0.133
SW	**0.888**	−0.135	−0.136
SFM	**0.854**	−0.137	−0.215
SDM	**0.922**	−0.091	−0.201
SMC	**0.906**	−0.311	0.046
PL	0.193	−0.164	**0.850**
PW	−0.163	**0.650**	**−0.672**
FL	0.564	**0.707**	0.274
FW	0.438	**0.805**	0.272
FM	0.268	**0.818**	0.096
Eigenvalue	4.723	2.468	1.457
Variability (%)	47.233	24.681	14.574
Cumulative (%)	47.233	71.915	86.488

PC1–3: principal components 1–3. Values ˃ 0.6 (in bold) are significant. Fruit and peduncle traits at 0, 1, 2, 3, 4, 5, 7, 14, 21 and 28 days after anthesis (DAA); shoot traits at 42, 49, 56 and 63 days after sowing (DAS); SL—shoot length (mm); SW—shoot width (mm); SFM—shoot fresh mass (g); SDM—shoot dry mass (g); SMC—shoot moisture content (%); PL—peduncle length (mm); PW—peduncle width (mm); FL—fruit length (mm); FW—fruit width (mm); FM—fruit mass (kg).

**Table 7 plants-12-00532-t007:** Genetic parameters for shoots, intact and harvested peduncles, and fruit traits of *L. siceraria* landraces at different growth stages.

Variables	δ^2^g	δ^2^e	δ^2^p	GM	PCV	GCV	ECV%	h^2^	GA
SL	7296.5	4156.0	11452.5	312.1	60.6	48.4	36.5	36.3	0.4
SW	0.1	0.09	0.14	0.7	4.5	2.8	3.5	60.5	0.3
SFM	47.4	35.0	82.3	16.4	22.4	17.0	14.6	42.5	0.4
SDM	0.4	0.3	0.6	1.8	5.8	4.4	3.7	41.9	0.4
SMC	342.5	6.9	349.5	83.8	20.4	20.2	2.9	2.0	0.5
PLi	5.2	1.8	7.0	4.1	13.1	11.3	6.6	25.6	0.4
PWi	3.1 × 10^−2^	5.6 × 10^−3^	3.7 × 10^−2^	0.4	3.0	2.7	1.1	15.0	0.4
FLi	3.4	1.9	5.2	3.3	12.6	10.1	7.5	35.5	0.4
FWi	1.2	0.6	1.8	2.0	9.6	7.9	5.5	32.5	0.4
PLh	177.2	218.1	395.3	61.9	25.3	16.9	18.8	55.2	0.3
PWh	1.6 × 10^−2^	4.2 × 10^−2^	2.0 × 10^−2^	0.7	1.7	1.5	0.8	21.2	0.4
FLh	7771.7	639.4	8411.1	193.9	65.9	63.3	18.2	7.6	0.5
FWh	676.1	84.0	760.1	87.8	29.4	27.7	9.8	11.1	0.5
FMh	0.2	1.7 × 10^−2^	0.2	0.6	5.5	5.2	1.7	10.0	0.5

Variables: δ^2^g—genotypic variance; δ^2^e—environmental variance; δ^2^p—phenotypic variance; GM—grand mean; PCV—phenotypic coefficient of variation; GCV—genotypic coefficient of variation; ECV%—environmental coefficient of variation; h^2^—broad sense heritability; GA—genetic advancement. Variables: SL—shoot length (mm); SW—shoot width (mm); SFM—shoot fresh mass (g); SDM—shoot dry mass (g); SMC—shoot moisture content (%); PLi—peduncle length of intact fruits (mm); PWi—peduncle width of intact fruits (mm); FLi—fruit length of intact fruits (mm); FWi—fruit width of intact fruits (mm); PLh—peduncle length of harvested fruits (mm); PWh—peduncle width of harvested fruits (mm); FLh—fruit length of harvested fruits (mm); FWh—fruit width of harvested fruits (mm); FMh—fruit mass of harvested fruits (kg).

## Data Availability

The research data can be requested from the authors.

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
