# Peer review of "Variation in Shoot, Peduncle and Fruit Growth of Lagenaria siceraria Landraces"

_plants, 2023, doi:10.3390/plants12030532_

Round 1

Reviewer 1 Report

A file is submitted.

Reviewer 2 Report

This is an interesting study. However, it is difficult not to get lost in the huge amount of raw data. An interesting aspect is, that the peduncles are obviously contractile. This seems to be hitherto only known in detail for the roots of some monocots. It is therefore desirable to describe the anatomical changes leading to contraction a bit more detailed. In the discussion the author writes a lot about the effect of phytohormons. This is partly trivial and partly just speculation and should be drastically shortened or even removed.  Instead there should be a clearer focus on the meaning of the data in the context of the research problem instead of discussing the data as such. This requires probably also a more clear cut and goal centered structure of the introduction.
